# Tomographic mapping of the hidden dimension in quasi-particle interference

C. A. Marques [1,5], M. S. Bahramy [2,5 ✉], C. Trainer[1], I. Marković[1,3], M. D. Watson [1], F. Mazzola[1], A. Rajan [1], T. D. Raub [4], P. D. C. King [1] & P. Wahl [1 ✉]

Quasiparticle interference (QPI) imaging is well established to study the low-energy electronic structure in strongly correlated electron materials with unrivalled energy resolution. Yet, being a surface-sensitive technique, the interpretation of QPI only works well for anisotropic materials, where the dispersion in the direction perpendicular to the surface can be neglected and the quasiparticle interference is dominated by a quasi-2D electronic structure. Here, we explore QPI imaging of galena, a material with an electronic structure that does not exhibit pronounced anisotropy. We find that the quasiparticle interference signal is dominated by scattering vectors which are parallel to the surface plane however originate from bias-dependent cuts of the 3D electronic structure. We develop a formalism for the theoretical description of the QPI signal and demonstrate how this quasiparticle tomography can be used to obtain information about the 3D electronic structure and orbital character of the bands.

[1] SUPA, School of Physics and Astronomy, University of St Andrews, North Haugh, St Andrews, Fife KY16 9SS, UK. [2] Department of Physics and Astronomy, The University of Manchester, Oxford Road, Manchester M13 9PL, UK. [3] Max Planck Institute for Chemical Physics of Solids, Nöthnitzer Strasse 40, 01187 Dresden, Germany. [4] School of Earth and Environmental Sciences, University of St Andrews, Irvine Building, St Andrews KY16 9AL, UK. [5] These authors contributed equally: C. A. Marques, M. S. Bahramy. ✉email: m.saeed.bahramy@manchester.ac.uk; wahl@st-andrews.ac.uk

Quasiparticle interference (QPI) has in recent years developed into a powerful tool for the characterization of electronic structure. Due to its unrivalled energy resolution, it has been highly successful in unravelling the detailed dispersion relation and structure of the superconducting order parameter in many correlated electron materials[1–3]. It is, however, inherently limited by imaging of the electronic states in two dimensions, and has consequently predominantly been applied to materials where the electronic structure can be approximated as two-dimensional, either due to a strong anisotropy, e.g., in layered materials[1,2], or because the states are purely two-dimensional, as is the case for surface states[4,5].

In a material with a crystal structure with high symmetry, for example, cubic, the interpretation of QPI will contain contributions from the full three-dimensional electronic structure, and scattering from defects that are below the surface layer will become non-negligible[6]. This makes the interpretation of the QPI patterns challenging as illustrated in Fig. 1a: a 2D measurement of the density of states at the surface contains information from a three-dimensional band structure and distribution of defects.

To describe how the quasiparticle interference measured at the surface layer relates to the bulk electronic structure requires addressing what the correct mapping between surface and bulk reciprocal spaces for its description is. Figure 1b illustrates this point for a rocksalt crystal structure: the primitive cell in the bulk cannot be used to describe a (001) surface. The minimal unit cell required to describe the surface as well as its electronic structure is thus larger. As a consequence, the surface BZ in which the electronic structure is measured by QPI is not just a cut or projection of the bulk BZ, but a folded version of it.

In Fig. 1c, we show a Fermi surface in the bulk Brillouin zone and in the minimal Brillouin zone required to describe the surface (Fig. 1d). In many face-centred cubic systems, the Fermi surface has pockets at the zone boundary of the bulk BZ, as sketched in Fig. 1c. Considering a constant $k_z$ plane, the folding in the surface layer maps out-of-plane scattering vectors between hot spots on the Fermi surface into the same $k_z$-plane (Fig. 1e, f).

A material with these properties is the mineral galena (PbS). It naturally crystallizes in a rock-salt crystal structure and exhibits a perfect cleaving plane along the {100}-directions, thus providing a 3D band structure with cubic symmetry and atomically flat surfaces. It is a narrow direct bandgap semiconductor, whose properties differ from those of more conventional semiconductors (e.g., GaAs), where the bandgap occurs at the Γ-point in the centre of the Brillouin zone. In galena, the bandgap is at the L-point at the boundary of the first Brillouin zone (compare Fig. 2a) and its width decreases with decreasing temperature[7,8] as well as under pressure[9]. The electronic structure is well known and features particle−hole symmetry across its semiconducting gap[10]. It undergoes topological changes as a function of energy when moving away from the valence band maximum (VBM) and from the conduction band minimum (CBm) with increasing absolute energy. At the VBM and CBm, there are no states in the $k_z = 0$ plane but only at the edges of the folded BZ (analogous to

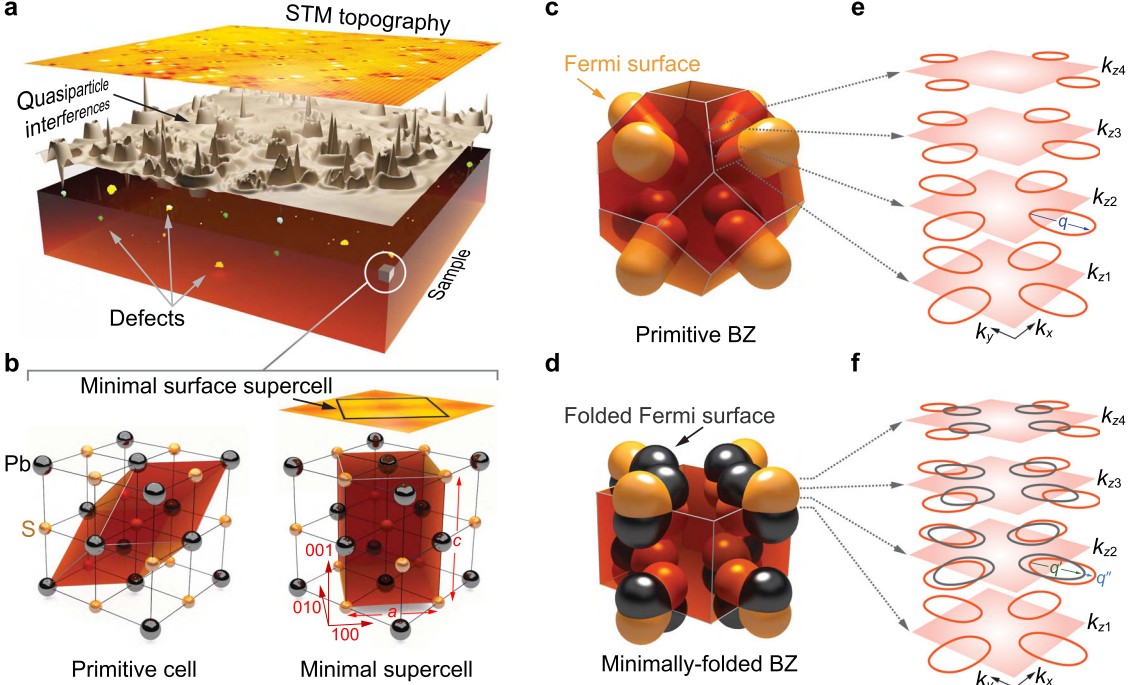

**Fig. 1 Crystal structure and Brillouin zone of PbS. a** Visualization of quasiparticle interference in an isotropic material. A typical STM topography (obtained from PbS), exhibiting an atomically flat surface with a wide range of defects, is shown ($V_{set} = 0.8$ V, $I_{set} = 0.2$ nA). The (schematic) differential conductance map, which is proportional to the density of states, shows clear signatures of quasiparticle interference (QPI), evidenced as oscillatory patterns around defects in the surface, as well as sub-surface layers. These patterns contain information from the full electronic structure. **b** Difference between the primitive cell of the bulk and surface. Left: the primitive cell of the bulk FCC crystal structure shown in a unit cell of PbS. Right: minimal supercell with lattice parameters $a$ and $c = \sqrt{2}a$ required to describe the surface. All crystallographic directions in the following refer to this supercell. **c** Schematic bulk Brillouin zone (BZ) of a cubic material with Fermi pockets centred at the zone boundaries, corresponding to the primitive cell shown on the left in **b** with a schematic electronic structure of an FCC material. The constant energy surface close to the top of the valence band in PbS is represented by pockets centred at the L-point (centre of the hexagonal faces). There are no states in the $k_z = 0$ plane. **d** Brillouin zone corresponding to the minimal supercell shown on the right in (**b**), resulting in folding of the Brillouin zone and hence of the Fermi surface. **e** Cuts at different $k_z$ planes for the Fermi surface in (**c**). With decreasing $k_z$ the size of the contours increases. There is one dominant intra-pocket scattering vector **q**. **f** Decreasing $k_z$ planes for the Brillouin zone shown in (**d**). Due to the folding, there are two new intra-pocket scattering vectors, **q**′ and **q**″, that do not exist in the bulk Brillouin zone.

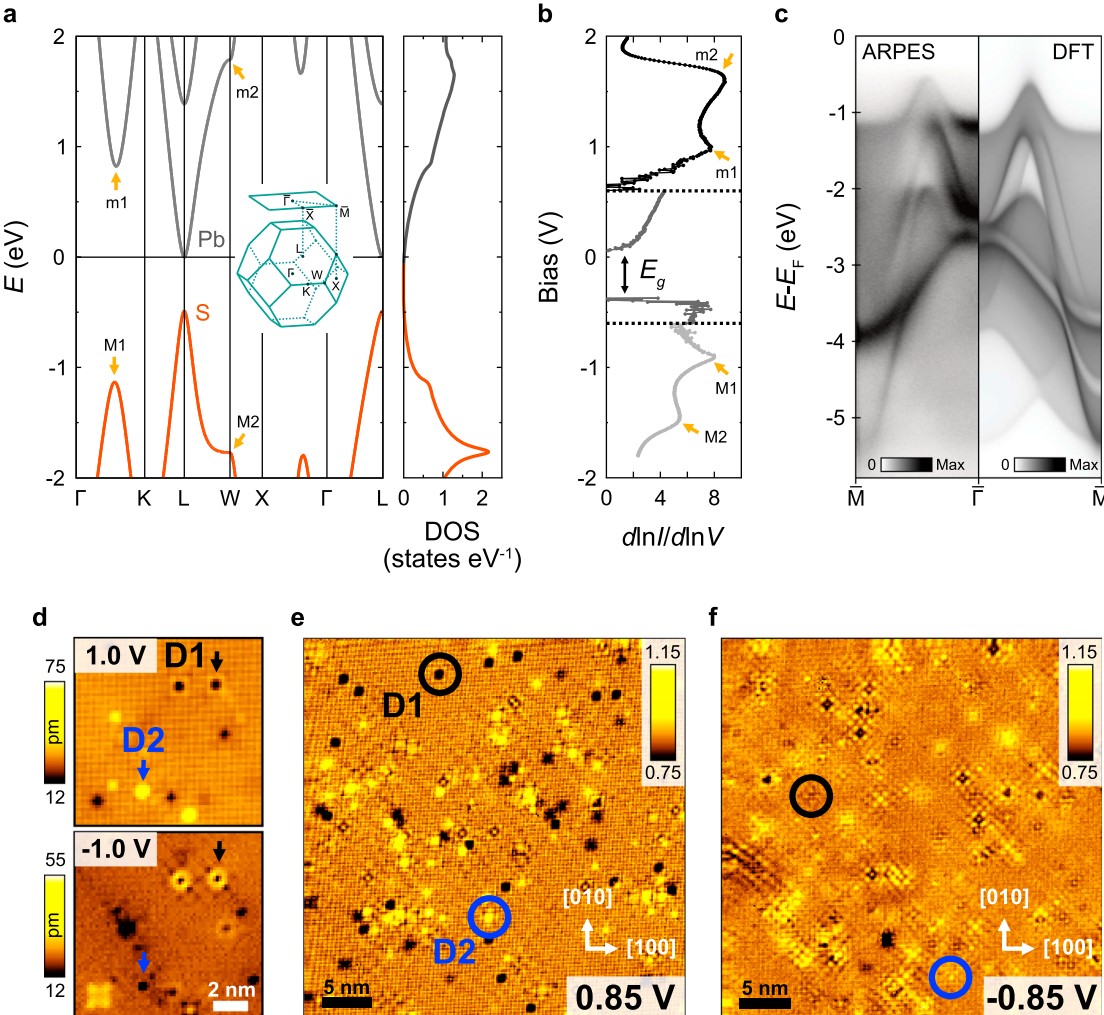

**Fig. 2 Band structure and density of states. a** Band structure of PbS from a tight-binding model for the bulk FCC BZ is shown on the left. The valence band is made up of sulfur $p$-orbitals, while the conduction band is mainly due to lead $p$-orbitals. The direct bandgap occurs at the $L$-point. Van-Hove singularities are indicated by M1 and M2 for the valence band and m1 and m2 for the conduction band. The panel on the right shows the total density of states. **b** Tunnelling spectroscopy, $d \ln I/d \ln V$, of galena, showing the gap $E_g$ between conduction and valence band. A gap of 0.34 V is observed, with the bottom of the conduction band pinned to the Fermi level. Four peaks can be observed, which can be identified with the van Hove singularities in the band structure. Because of the strong increase in density of states at positive bias voltages, spectra have been recorded in three energy ranges indicated by dashed lines, which we probe with different setpoint conditions (Black: $V_{set} = 1.2$ V, $I_{set} = 0.1$ nA; dark grey: $V_{set} = 0.6$ V, $I_{set} = 0.2$ nA; light grey: $V_{set} = 0.8$ V, $I_{set} = 0.2$ nA). **c** Electronic band structure along the $\overline{\Gamma} - \overline{M}$ direction of the surface-projected Brillouin zone as measured by ARPES (left) and calculated by DFT (right), showing excellent agreement. The DFT calculation simulates the finite $k_z$-resolution of the ARPES experiment using empirical parameters (see Supplementary Note 8). **d** Topographic images showing characteristic defects at lead (D1) and sulfur (D2) sites for positive and negative bias (top: $V_{set} = 1$ V, $I_{set} = 150$ pA; bottom: $V_{set} = -1$ V, $I_{set} = 100$ pA). **e** Spatial map of $d \ln I/d \ln V$ at 0.85 V and **f** at $-0.85$ V. The circles identify the same types of defects as in (**d**).

Fig. 1d), making this material an ideal choice to establish the contribution from states with non-zero $k_z$ to the quasiparticle interference. It is only at about 0.6 eV below (0.8 eV above) the VBM (CBm) that states appear in the $k_z = 0$ plane (compare Fig. 2a). From angular-resolved photoemission, there is no evidence for a surface state in the vicinity of the Fermi energy (Fig. 2c, also ref. [8]), and so the detected QPI signal originates solely from the bulk electronic structure.

The interpretation of the quasiparticle interference detected at the surface of a material without a clear 2D anisotropy thus immediately raises several fundamental questions: (1) what is the effect of the three-dimensionality of the electronic structure on the quasiparticle interference detected at the surface? (2) How does the zone-folding from the bulk Brillouin zone (BZ) to the minimal cell required to describe the surface affect the scattering

processes contributing to the QPI? and (3), how do sub-surface impurities contribute to the scattering signal?

Here, we address these questions through imaging of quasiparticle interference at the surface of the mineral galena (PbS), which has a bulk electronic structure with cubic symmetry. Comparison with theory reveals that in galena, the QPI signal observed at any given bias voltage is dominated by hotspots in certain $k_z$ planes of the bulk BZ, demonstrating 3D quasiparticle tomography of the electronic structure.

## RESULTS

The galena sample was cleaved in-situ at cryogenic temperatures, producing a high-quality atomically-flat surface. A representative topographic STM image, measured at 20 K, shows atomic resolution, exhibiting the square atomic lattice of the lead atoms

(Fig. 1a and Supplementary Fig. 1a). The surface exhibits a rich variety of defects found at both lead and sulfur lattice sites as a consequence of its natural origin and aiding in the observation of quasiparticle interference. In excellent agreement with the calculated total density of states (Fig. 2a) and with the previous reports[8,11,12], typical tunnelling spectra (Fig. 2b) reveal the energy gap of galena on the order of $E_g \approx 0.34$ eV. We find that the CBm is pinned to the Fermi level, indicating $n$-doping of the sample. Additionally, two sets of peaks are observed in tunnelling spectra: M1 and M2 at negative energies and m1 and m2 at positive energies, associated with van Hove singularities in the band structure related to band minima (m) and maxima (M) (compare Fig. 2a) in agreement with the known density of states[13,14]. Comparison of the calculated band structure with ARPES measurements of the electronic structure show excellent agreement between the two (Fig. 2c).

In the rock salt crystal structure of PbS, defects at sulfur and lead sites are expected to lead to significantly different scattering patterns. Figure 2d shows characteristic defects on the lead and sulfur sites imaged at positive and negative bias voltage with a distinctly different appearance. From the chemical analysis of the sample (see Supplementary Note 1), the predominant defects are Bi ($n$-doping) and, with a smaller concentration, Ag ($p$-doping), leading to a net $n$-doping of the sample. Spatially-resolved maps of $d \ln I / d \ln V(\mathbf{r}, V_b)$ at $V_b = 0.85$ V and $-0.85$ V are shown in Fig. 2e, f, respectively. The same type of defects as in Fig. 2d are highlighted on both maps. At both energies, clear QPI patterns are visible as spatial modulations around all defects, which appear in the Fourier transformation (Fig. 3a, e) as characteristic wave vectors along the [100] and [110] directions. The energies shown in Fig. 3a, e are close to the VBM and the CBm, respectively, where the constant energy surfaces have no states in the $k_z = 0$ plane, yet we see strong signatures of quasiparticle scattering. Comparison with the bulk electronic structure reveals that the scattering we observe along the [100] direction can only originate from scattering between disconnected hole (electron)-pockets around the L-point at the boundary of the bulk Brillouin zone, as represented in Fig. 1c, d. The same pockets account for the scattering vectors visible along the [110] direction. Moving further away from the gap edge, these pockets become connected, as observed in Fig. 3g. The Fourier transformations show closed patterns whose characteristic wave vector decreases with increasing absolute energy.

In order to understand the origin of the features in QPI, we have developed a formalism based on $T$-matrix calculations to describe the QPI patterns from the full three-dimensional electronic structure. We use a tight-binding model for the bulk obtained from DFT calculations to obtain the bare Green's function $G_0(\mathbf{k}, \omega)$ of the unperturbed system (see Methods section), and then evaluate the $\mathbf{q}$-resolved density of states $\rho(\mathbf{q}, \omega)$ from

$$\rho(\mathbf{q}, \omega) = -\frac{1}{2\pi i} \sum_{\mathbf{k}} \mathrm{Tr} \{G(\mathbf{k}, \mathbf{k} - \mathbf{q}, \omega) - G^*(\mathbf{k}, \mathbf{k} + \mathbf{q}, \omega)\}, \quad (1)$$

where $G(\mathbf{k}, \mathbf{k}', \omega)$ is the retarded Green's function of the system including the impurities (see Methods section). Note that here, unlike in the QPI calculations done usually, the wave vectors $\mathbf{k}$ and $\mathbf{k}'$ are three-dimensional wave vectors. We account for scattering processes through the $T$-matrix which encodes the details of the scattering potential $V_{\mathbf{k},\mathbf{k}'}$ (see Methods section, eq. (4)).

To obtain the surface-projected QPI, $\tilde{\rho}(q_x, q_y, \omega)$, we first divide the folded bulk BZ into a series of $(q_x, q_y)$ planes parallel to the surface, and for each plane calculate the bulk QPI $\rho(q_x, q_y, \omega)|_{k_z}$, using the Green's function defined in Eq. (1). While we calculate the QPI in planes with $q_z = 0$, scattering vectors with non-zero $q_z$

are accounted for implicitly through the folding. We then integrate over all $k_z$ components to obtain the QPI, $\tilde{\rho}(q_x, q_y, \omega)$. The resulting QPI pattern obtained from a calculation describing the $k_z$ dependence and the orbital character of the wave functions is shown in Fig. 3 next to the experimental data. To fully describe the experimental data, we manually adjust the ratio of the contributions from $p_{x,y}$, and $p_z$ orbitals. The extracted ratio $p_z/p_{x,y}$ is shown in Supplementary Fig. 8. This change in orbital contribution with bias voltage reflects the diversity of scattering processes from different defects in the bulk and their directional anisotropy and energy dependence. The comparison reveals excellent and nearly quantitative agreement.

Our analysis reveals several important points: (1) For both positive and negative bias voltages, $p_z$- and $\{p_x, p_y\}$-orbitals create distinct QPI patterns, despite the orbital degeneracy of bulk electronic bands. (2) As a function of bias voltage, we observe a change in the relative intensity with which bands with $p_z$- and $\{p_x, p_y\}$-orbital character appear in the observed QPI: the ratio is found to continuously increase with increasing energy both above and below the Fermi energy. (3) The experimental QPI pattern can only be reproduced when accounting for the full $k_z$-dependence of the electronic structure. A quasi-2D model would fail spectacularly for the present system.

## DISCUSSION
Our framework to describe the QPI allows us to provide an understanding of these observations and disentangle the contributions from different $k_z$ planes to make the connection from the QPI patterns to the bulk electronic structure.

To better understand the energy dependence of the relative contributions of bands with different orbital character to the observed QPI patterns, a clear understanding of symmetry and dimensionality of the electronic bands and their interaction with the existing impurities is necessary. Bulk PbS has an isometric crystal structure with the highest possible symmetry. As such, the $p$-orbital manifold is subject to an isotropic cubic crystal field, protecting the orbital degeneracy of the energy bands. At the surface, this is no longer the case, due to the surface discontinuity, suppressing the electron hopping along the surface normal. The (001) surface termination creates a tetragonal crystal field near the surface, causing an energy splitting between the $p_z$ and $\{p_x, p_y\}$ orbitals. As the latter are spatially distributed along [100] and [010] directions, they contribute predominantly to the QPI features at the zone centre or along the corresponding in-plane surface reciprocal wave vectors, [10] and [01] (see Fig. 3). On the other hand, the $p_z$ orbital, due to its apical distribution, can isotropically contribute to scattering in any in-plane direction. As a result, it becomes the main channel for impurity scattering within and near the [110] plane, appearing as diagonal features in the observed and calculated QPI. Furthermore, both Pb-$p_z$ and S-$p_z$ can strongly interact with the impurities beneath the surface, as they are the only orbitals that can form strong interlayer $\sigma$-type bonding; $\{p_x, p_y\}$ can only form $\pi$-type bonding between the PbS layers, which is much weaker than $\sigma$ bonding, leading to weaker scattering patterns.

Such a characteristic difference between these two orbital sets and their contribution to the QPI reveals a fundamental distinction in the orbital character of the underlying scattering processes. To further explain how they respond to the bias variation, we also need to consider the energetic alignment of the impurity states and their relative coupling to the host energy bands.

As shown in Fig. 3, at negative bias voltages close to the Fermi energy, both $p_z$ and $\{p_x, p_y\}$ contribute almost equally to the observed QPI, but the latter becomes increasingly dominant at

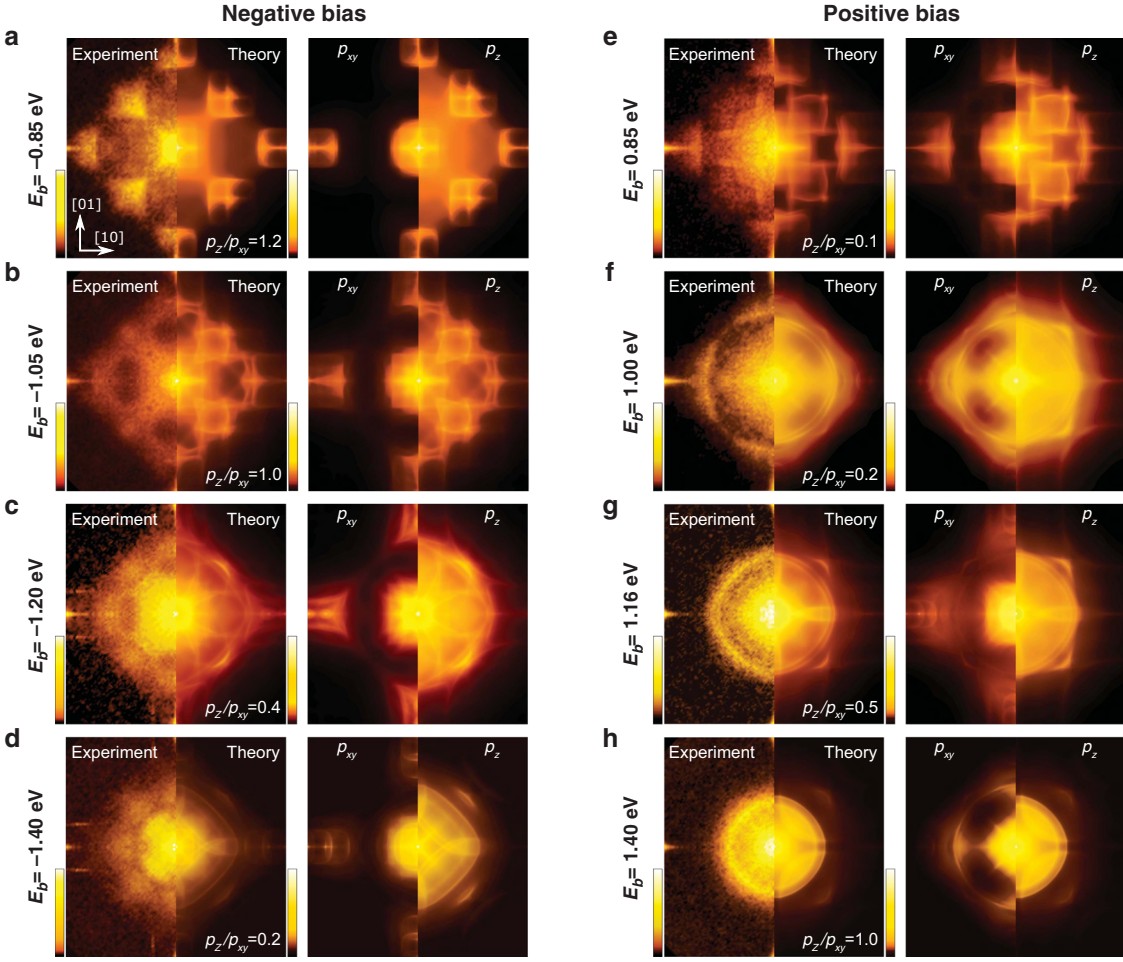

**Fig. 3 Comparison of experimental QPI patterns with _T_-matrix calculations. a−d** QPI patterns at negative energy with decreasing energy. The panels on the left show the comparison between the experimental data and the _T_-matrix calculation of the QPI for the full bulk electronic structure. For each energy, the ratio $p_z/p_{xy}$ of the orbital contributions required to describe the experimental data is indicated. The panels on the right show the QPI patterns originating from sulfur $\{p_x, p_y\}$ (left) and $p_z$ states (right). **e−h** QPI patterns at positive energies with increasing energy with the orbital contributions from the Pb _p_-orbitals. Left and right panels as for (**a−d**). Near the band edge, disconnected QPI patterns are observed as expected for the top and bottom of the valence and conduction bands, respectively, which join as the absolute energy increases. For all panels, the **q** vector spans the reciprocal space from $(\frac{-2\pi}{a}, \frac{-2\pi}{a})$ to $(\frac{2\pi}{a}, \frac{2\pi}{a})$, where _a_ is the lateral lattice parameter of the tetragonal supercell shown in Fig. 1b.

more negative $V_b$. This means that scattering in the $\{p_x, p_y\}$ channel at surface impurities becomes dominant at high negative bias voltages. Given that the existing impurities are mainly charge-donors (cationic), this finding is not unexpected. Cations tend to have impurity states at higher energies than anions. At negative bias voltages, we are probing anionic S-derived bands, further away from the impurity states, lowering the probability of scattering in total and from defects in the sub-surface layers in particular. Thus the resulting QPI tends to become more dominated by the surface $\{p_x, p_y\}$ orbitals as $V_b$ becomes more negative. For a positive $V_b$, the situation is the opposite. Since now both the impurity states and Pb-derived bands are cationic, near the band edge, we see a more significant contribution from $\{p_x, p_y\}$ orbitals. This can be interpreted as a higher probability of scattering from the top surface layer as the Pb-site impurities in this layer couple more strongly to the neighbouring Pb ions through $\{p_x, p_y\}$ orbitals. As $V_b$ increases, the scattering from defects in subsurface layers through $p_z$ orbitals appears to gain an increasing strength, leading to a dramatic enhancement of the ratio $p_z/p_{x,y}$. Such a dimensional anisotropy in quasiparticle scattering and its dependence on the chemical character of impurities underlines

the importance of our bulk-based analysis of the QPI patterns observed at the surface.

The relation between the $k_z$ planes dominating the QPI signal and the electronic structure is illustrated in Fig. 4. Figure 4a shows a cut in the (010) plane through constant energy surfaces of the electronic structure at increasing absolute energies $E_1…E_4$. It is obvious that the quasiparticle interference patterns near the band edge cannot originate from the $k_z = 0$ plane as there are no states—demonstrating the need for a description of the QPI that accounts for the full 3D band structure. For surface defects, which will make the dominant contribution, the scattering vectors that contribute most connect pockets of the band structure with group velocities in the surface plane, indicated by the red dashed line in Fig. 4a.[15] To identify the scattering vectors dominating the quasiparticle interference patterns and efficiently calculate the QPI signal, we evaluate the flatness of the combined pockets in the electronic structure in the minimal cells required for the surface (folded) and the bulk (primitive) Brillouin zone, Fig. 4b. In the folded zone, the points with in-plane group velocity map into a single $k_z$-plane, and hence the dominant scattering vectors have a $z$-component which is zero, making the calculation of the QPI

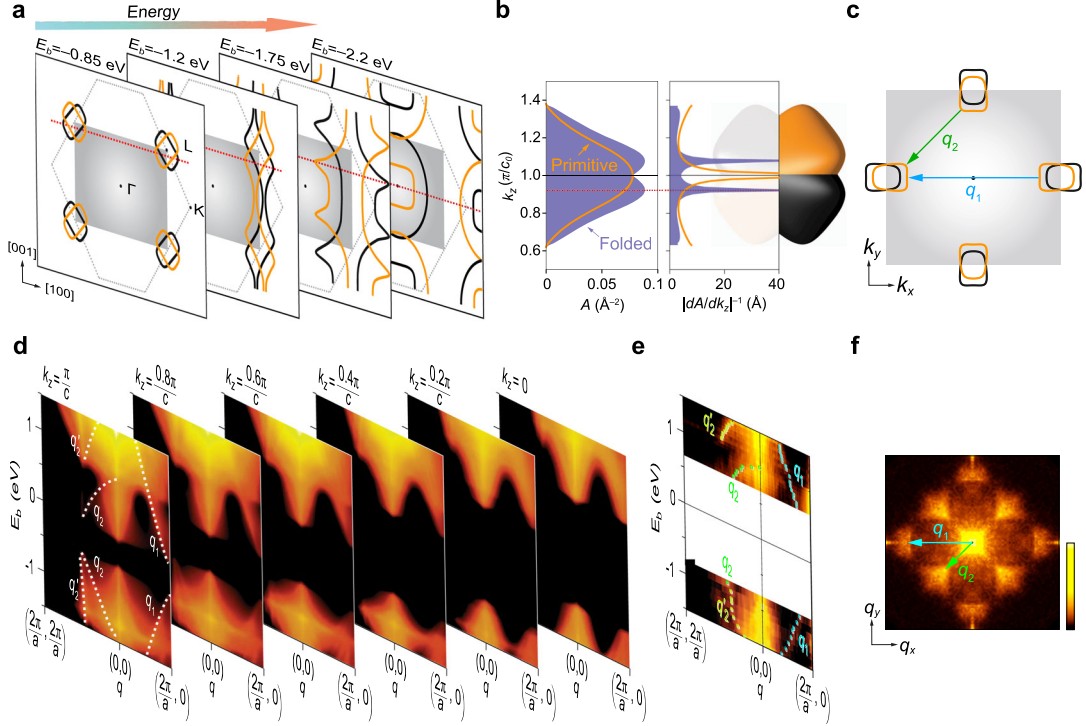

**Fig. 4 $k_z$ selectivity of quasiparticle interference. a** Representation of (010)-planes at $k_y = 0$ with increasing absolute energy. The first frame depicts an energy close to the VBM. Lines indicate constant energy contours of the bulk electronic structure, orange lines in the bulk Brillouin zone, and black lines of folded bands in the surface Brillouin zone. The red dashed lines indicate the $k_z$ plane where the bands exhibit an in-plane group velocity. The folding maps the scattering vectors connecting hot spots in different $k_z$ planes into the same $k_z$ plane. With increasing absolute energy (indicated by the arrow), the dominant $k_z$-plane for scattering moves towards $k_z = 0$. **b** To determine the dominant $k_z$ plane, we use the surface area $A$ (left panel, showing the merged area enclosed by the union of the two pockets for the folded zone) to determine the flatness $|dA/dk_z|^{-1}$ (right panel) as a function of $k_z$ for the pockets in (**a**) (shown here for $E = -0.85$ eV). Once folding is accounted for, the points with the largest flatness occur at $k_z = 0.88\pi/c$, indicated by the dashed red line. **c** Cut along the red line in (**a**) for the first layer. The vectors $\mathbf{q}_1$ and $\mathbf{q}_2$ connect unfolded Fermi surfaces. **d** Maps of the **q**-resolved density of states $\tilde{\rho}(\mathbf{q}, E)$ due to QPI, taking into account the energy dependence of the orbital contributions, as a function of energy $E$ for different $k_z$ planes. **e** Corresponding experimental data along the same path as in (**d**). The circles indicate the position of the peaks (error bars represent **q**-space resolution of the d$I$/d$V$-maps). **f** QPI layer at $V_b = -0.85$ V, with the arrows as in (**c**).

signal more efficient. To determine the dominant $k_z$-plane, we define the flatness as the inverse of the derivative of the area with respect to $k_z$, i.e., $(dA/dk_z)^{-1}$. While using the bulk Brillouin zone shows a dominant $k_z$ vector at the zone boundary, in the folded Brillouin zone, corresponding to the minimal surface unit cell, the dominant $k_z$ plane is away from the high-symmetry planes— consistent with the experiment. The dominant $k_z$-plane extracted in this way is shown in Fig. 4c, with the resulting scattering vectors. We note that this dominant $k_z$ plane is energy-dependent, as the band pockets grow, the dominant $k_z$ plane moves towards the zone centre (see Fig. 4a). The flatness curve in Fig. 4b shows additional maxima at its edges which are of no significance in QPI calculations, as the Fermi surfaces at these $k_z$ cuts have a negligible cross-sectional area.

The quasiparticle dispersion obtained from the calculation for different $k_z$ planes is shown in Fig. 4d together with the experimentally observed one (Fig. 4e). The dispersion at energies close to the VBM and CBm is visible only at $k_z$ planes close to the zone boundary, near $k_z = \pi/c$, whereas planes close to $k_z = 0$ become more relevant for higher absolute energies. This becomes apparent from comparing the scattering vectors obtained from the experiment with the calculation (white dotted lines in Fig. 4d), showing agreement for some scattering vectors for $k_z = \frac{\pi}{c}$ ($q_1$ and $q_2$), whereas other vectors clearly do not fit ($q_2'$) but are better described by scattering from a different $k_z$-plane (see Supplementary Fig. 5 for detailed comparison). The dominant scattering

vectors in the $k_z$ plane with the largest contribution at $-0.85$ V are shown in Fig. 4f superimposed to the experimental quasiparticle interference.

Our results highlight how quasiparticle interference imaging can be used to probe the three-dimensional electronic structure. As a basis, we have chosen a system that has a cubic symmetry and a three-dimensional electronic structure, and does not exhibit electron correlations, which enables for a complete and accurate theoretical description of the electronic structure and the wave functions, making this an ideal test system. The high precision of the band structure calculation is seen in the comparison with the ARPES data.

Different to quasi-two-dimensional materials, where it is often sufficient to only consider the $k_z = 0$ plane, in this system, the dispersion in the direction normal to the surface cannot be ignored: the QPI patterns observed in STM cannot be simply explained by the quasiparticle signal coming from a particular $k_z$-plane, or even just the $k_z = 0$ plane. What we rather find is that the QPI signal is dominated by bias-dependent $k_z$ planes, determined by the plane of dominant in-plane group velocity. Taking into account the surface Brillouin zone, we can understand the QPI signal once the correct $k_z$-plane is determined, resulting in excellent agreement between theory and experiment of both the dominant QPI features as well as their dispersion.

The picture that emerges has far-reaching consequences for the interpretation of quasiparticle interference in materials where the

electronic structure is not quasi-2D. In these cases, the dominant scattering plane, which can be multiple planes in multi-band systems, needs to be determined to be able to extract information about the electronic structure from the quasiparticle interference patterns. We note that in many anisotropic systems, the quasiparticle interference will be dominated by scattering vectors in high-symmetry planes, where the group velocity has to be in-plane by symmetry, for example $k$-vectors in the $k_z = 0$ plane and at the zone face[16]. The present example, however, shows that this is not generally true, and the dominant $k_z$ plane can differ from high symmetry planes and even become energy-dependent. Correct identification of the dominant $k_z$-planes and their energy dependence enables tomographic characterization of the bulk electronic structure from quasiparticle interference imaging.

In conclusion, we have demonstrated quasiparticle tomography of the bulk electronic structure from a surface-sensitive measurement. Modelling the scattering vectors due to the three-dimensional electronic structure using the zone folding that occurs in the surface region, we find that the dominant contribution to the quasiparticle interference imaging comes from an energy-dependent $k_z$-plane. We introduce a theoretical framework to efficiently compute the resulting QPI patterns. By accounting for the full 3D electronic structure and the orbital composition of the surface electronic bands, we are able to fully reconstruct the experimentally observed quasiparticle interference patterns from theory. Our results introduce a method to extract information about the 3D bulk electronic structure from a quasiparticle interference measurement in 2D.

## METHODS

**Sample details**. The sample used in this study is a mineral sample of galena collected in Malawi. Galena is the natural ore mined for lead and silver. Our sample is an argentiferous galena sample, i.e., has significant silver content. We have analyzed the sample composition by Inductively Coupled Plasma Mass Spectrometry (ICPMS), for details of the analysis see Supplementary Note 1. Galena has a cubic crystal symmetry with well-defined cleavage planes in the [001] directions, exposing mirror-like surfaces.

**STM measurements**. We performed STM measurements in a home-built ultra-low temperature STM, mounted in a dilution refrigerator[17], working in a cryogenic vacuum. The STM tip was made of Pt−Ir wire, prepared prior to measurements on a single crystal of Au(111) by field emission. The PbS sample was cleaved in situ at low temperatures. All measurements were performed at 20 K to thermally excite a sufficiently large number of charge carriers so that STM imaging becomes possible. The temperature was stabilized by a heater on the STM head. The bias voltage was applied to the sample, with the tip at virtual ground. Tunnelling spectroscopy was performed using a lock-in amplifier to measure differential conductance, d$I$/d$V$, where the bias voltage $V$ was modulated at a frequency of $\nu = 437$Hz. Due to the increase in differential conductance at large bias voltages, different regions of the measured bias voltages had to be probed with different bias and current setpoints, $V_{set}$ and $I_{set}$, so that both the semiconducting gap and the features at negative bias could be resolved. In this way, the measured bias was split into three intervals: $[-1.8, -0.6]$, $[-0.6, 0.6]$, and $[0.6, 2.0]$, in volts, as shown in Fig. 2b. The spectroscopic and QPI data shown in this work correspond to[18]

$$\frac{d\ln I}{d\ln V}(V) = \frac{(dI/dV)(V)}{I(V)/V}, \tag{2}$$

to allow consistent comparison between measurements taken with different bias and current setpoints.

**Angle-resolved photoemission spectroscopy**. We performed angle-resolved photoemission spectroscopy (ARPES) on the same sample, at the APE beamline of the Elettra synchrotron in Trieste, Italy[19]. The samples were cleaved in situ and measured at 78 K, using $p$-polarized synchrotron light with the photon energy of 70 eV, and the data were collected using a Scienta-Omicron DA30 electron energy analyzer.

**Calculations**. The QPI in momentum space, i.e., Eq. (1), was computed using the retarded Green's function $G$ defined as

$$G(\mathbf{k}, \mathbf{k}', \omega) = G_0(\mathbf{k}, \omega)\delta_{\mathbf{k},\mathbf{k}'} + G_0(\mathbf{k}, \omega)T_{\mathbf{k},\mathbf{k}'}(\omega)G_0(\mathbf{k}', \omega), \tag{3}$$

with $T$ being a matrix encompassing all scattering processes between the wave-

vectors $\mathbf{k}$ and $\mathbf{k}' = \mathbf{k} - \mathbf{q}$ by the impurity potential $V_{\mathbf{k},\mathbf{k}'}$,

$$T_{\mathbf{k},\mathbf{k}'}(\omega) = V_{\mathbf{k},\mathbf{k}'} + \sum_{\mathbf{k}''} V_{\mathbf{k},\mathbf{k}''} G_0(\mathbf{k}'', \omega) T_{\mathbf{k}'',\mathbf{k}'}(\omega), \tag{4}$$

and $G_0(\mathbf{k}, \omega)$ being the bare Green's function

$$G_0(\mathbf{k}, \omega) = \sum_n \frac{|n, \mathbf{k}\rangle\langle n, \mathbf{k}|}{\omega + i\eta - \epsilon_n(\mathbf{k})}, \tag{5}$$

resulting from the Bloch eigenvalues $\epsilon_n(\mathbf{k})$ and eigenvectors $|n, \mathbf{k}\rangle$ with the broadening factor $\eta$.

Since the impurities in our PbS samples are expected to be non-magnetic and randomly distributed in the whole system, we neglect the spin- and momentum-dependence of the scattering potential and regarded it as $V_{\mathbf{k},\mathbf{k}'} = V_0 \mathbb{1}$, where $\mathbb{1}$ is the identity matrix and $V_0$ is a constant potential, here fixed at $V_0 = 0.1$ eV added to the on-site energy at the defect site. Similarly, the resulting $T$ matrix was assumed to be only a function of $\omega$ and independent of $\mathbf{k}$. We then used the convolution technique developed by Kohsaka et al.[20] to compute $G(\mathbf{k}, \mathbf{k} - \mathbf{q}, \omega)$ over a fine $600 \times 600$ $\mathbf{q}$-mesh, spanning the entire surface BZ. To further accelerate the calculations and, more importantly, to account for the vacuum overlap, the Bloch eigenvalues $\epsilon_n(\mathbf{k})$ and eigenvectors $|n, \mathbf{k}\rangle$ were extracted from a low-energy tight-binding (TB) model, via a set of atomic-orbital (AO)-like Wannier functions[21]. We used the Fourier representation of these Wannier functions to transform all the Green's functions from their periodic lattice model to a continuum model[22–24]. The resulting QPI was subsequently projected onto a plane with a fixed height $z = 5$ Å, corresponding to the distance between the STM tip and the surface (in our model this corresponds to a plane 5 Å above the topmost PbS layer in the minimal bulk supercell).

To construct the TB model, we first performed a fully-relativistic DFT calculation using Perdew−Burke−Ernzerhof exchange-correlation functional as implemented in WIEN2K[25], for the tetragonal bulk supercell containing two formula units of PbS shown in Fig. 1b. The corresponding BZ was sampled by $20 \times 20 \times 16$ $\mathbf{k}$-points. To accurately account for the correlation effects, an additional non-local potential term was added to the DFT Kohn-Sham Hamiltonian using the modified Becke−Johnson scheme[26]. From this, we then downfolded a $12 \times 12$ TB model using twelve AO-like Wannier functions[21], representing the top six valence bands, dominated by S-2$p$ orbitals, and bottom six conduction bands, mainly of Pb-6$p$ character. The resulting TB Hamiltonian,

$$H = \sum_{\mathbf{R},\mathbf{R}'} t_{\mathbf{R},\mathbf{R}'}^{\mu\nu,\sigma\sigma'} c_{\mathbf{R}\mu\sigma}^\dagger c_{\mathbf{R}'\nu\sigma'} \tag{6}$$

describes the creation of an electron by the operator $c_{\mathbf{R}\mu\sigma}^\dagger$ at the lattice site $\mathbf{R}$ with the respective AO and spin characters $\mu$ and $\sigma$, after the annihilation of another electron with AO and spin characters $\nu$ and $\sigma'$ by the operator $c_{\mathbf{R}'\nu\sigma'}$ at site $\mathbf{R}'$. The corresponding hopping parameter is denoted by $t_{\mathbf{R},\mathbf{R}'}^{\mu\nu,\sigma\sigma'}$. Here, $\mu$ and $\nu$ can be any of the outermost $\{p_x, p_y, p_z\}$ orbitals of Pb and S.

By aligning the $c$-axis of the TB supercell along the crystalline [001] direction, we can ensure that all surface projections occur within a square superlattice exactly as found in the STM experiment. This results in a double folding of the wave functions in $k$-space along the $k_z$ direction, as shown in Fig. 1c, d. Correspondingly, any $k_z$ cut of the Fermi surface contains the energy contours from the primitive cell as well as their folded counterparts. An important outcome of such folding is that it enables us to selectively determine all major inter- and intra-pocket scattering vectors $\mathbf{q}$ without knowledge of their $q_z$ component (see Fig. 1e, f). One can identify which $k_z$ component of bulk wave functions contribute the most to the QPI pattern observed at the surface and how it evolves by varying the bias voltage. Also, since in our methodology we use AO-like Wannier functions, we can easily decompose the calculated QPI in terms of different orbital components. As such, it is possible to specify how strongly each orbital contributes to the observed QPI.

## Data availability
Underpinning data will be made available at ref. [27].

## Code availability
The computational data and source code are available upon reasonable request to M.S. Bahramy (m.saeed.bahramy@manchester.ac.uk).

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

## Acknowledgements

C.A.M. acknowledges funding from EPSRC through EP/L015110/1, and C.T. and P.W. through EP/R031924/1. P.W. and T.R. are grateful for support through SARIRF funding by the University of St Andrews. I.M. acknowledges studentship support through the International Max Planck Research School for Chemistry and Physics of Quantum Materials. M.D.W., A.R., and P.D.C.K. acknowledge funding from The Leverhulme Trust. F.M. and P.D.C.K. acknowledge funding from the European Research Council (through the ERC-714193-QUESTDO project). We thank Elettra synchrotron for access to the APE-LE beamline, which contributed to the results presented here, and we gratefully acknowledge Chiara Bigi and Ivana Vobornik for technical assistance. F.M. and P.D.C.K. are grateful for support by the project CALIPSOplus under Grant Agreement 730872 from the EU Framework Programme for Research and Innovation HORIZON 2020. P.D.C.K. acknowledges support from the UK Royal Society.

## Author contributions

M.S.B. and P.W. conceived and led the project. C.A.M. performed STM measurements with assistance from C.T.; C.A.M. analyzed the STM data; T.R. provided and characterized the sample; I.M., M.D.W., F.M., A.R., and P.D.C.K. performed ARPES measurements and analyzed the ARPES data; M.S.B. developed the quasi-particle formalism for 3D QPI and did the calculations. C.A.M. and M.S.B. prepared the figures. P.W., C.A.M., and M.S.B. wrote the manuscript with contributions from all authors.

## Competing interests

The authors declare no competing interests.
