## [Peer Review File · Nature Communications]

REVIEWER COMMENTS

Reviewer #1 (Remarks to the Author):

In "Tomographic mapping of the hidden dimension in quasiparticle interference" by Marques et al., the authors present a method for analyzing quasiparticle interference images, which permits extraction of information about the bulk electronic structure of galena, a fully three-dimensional--rather than quasi-two dimensional--material. They develop a method based on T-matrix calculations for extracting this information, in order to go beyond past work limited to study of quasi-two dimensional systems. Experimental results are supported by density functional theory and tight-binding theoretical methods to demonstrate the effectiveness of their approach.

The work serves as a useful step in studying truly three dimensional electronic structures, showing excellent agreement between experiment and theory. Given the widespread use of quasiparticle interference imaging in study of condensed matter systems, this proof-of-concept work addresses an important issue in extending this experimental technique to a broader class of materials. I therefore think is suitable for publication in Nature Communications. I do ask, however, for the authors to clarify how they determined the ratios of p_z and $p_{\{x,y\}}$ orbitals used to match to experiment for Fig. 3. Was this ratio determined manually and by eye, or was it automated and achieved by a quantitative fit? If this was achieved manually, the authors should acknowledge this, comment on the shortcomings or why it is good enough, and discuss how to make this more quantitative. If it was achieved quantitatively, explicit discussion of this approach should be added, again commenting on shortcomings and possible improvements.

Reviewer #2 (Remarks to the Author):

In this manuscript, Marques et al. investigated the quasiparticle interference (QPI) images on PbS(100) surface through scanning tunneling microscopy. By considering the folded band structures in minimal supercell, they found the QPI images at different bias are dominated by the scattering vectors in energy-dependent k_z -plane. Through T matrix calculations with full 3D electronic structure and orbital composition of surface band, the QPI patterns can be theoretically computed and accorded with experiments well. Their results suggest the 3D electronic structures have strong contribution in the QPI images, which can be used to extract the bulk band structures. This work is quite interesting, and the theoretical framework is thought to be convincing. However, before this manuscript is published, some revisions are needed. My comments and questions are as follow.

1. According to their theoretical framework, the QPI images are dominated by the energy-dependent k_z plane, which can be derived as shown in Fig.4 for $E_b=0.85\text{eV}$ for example. The calculated QPI patterns at different energies are shown in Fig.3, where however the dominating k_z planes for each energy were not given. The corresponding k_z planes at each energy which are used to calculate the QPI patterns in Figure 3 should also be presented in these figures in order for the authors to better follow their theoretical framework.

2. The authors show the scattering vectors connecting the pocket-like constant energy surface (CES) in Fig. 4c. Since the QPI patterns come from the energy-dependent k_z plane, the CES in different k_z plane should be quite different from others. I wonder how the scattering vectors connect the CES in other k_z planes.

3. The authors only show the dispersion of q_1 and q_3 in Fig. 4d. The scattering vectors q_2 and q_4 in folding BZ are missing. Why?

Reviewer #3 (Remarks to the Author):

Dear editorial office, dear authors,

I read the MS 'Tomographic mapping of the hidden dimension in quasi-particle interference' by Marques, Wahl et al.

The presented work is very impressive to me, it's worth to be presented to a broad audience.

Still, I feel that the material and the discussion need quite some improvements and changes following the questions raised below.

After major revisions, the manuscript might be suitable for publication.

As explained in very detail, the authors have found signatures of the bulk electronic structure on the surface of a crystal provided by the scattering at point-like impurities in the surface-near region. The density of states at the surface was mapped by differential conductance maps acquired by STM. So, long-range periodic changes of the density of states were found which are related to features of a perturbed bulk-like electronic structure which was derived by a Green's function technique. This relation of experimental and theoretical results is not very convincing since it is not clear how the

modified bulk electronic structure reflects the surface electronic structure, since the surface electronic structure was not considered explicitly.

By showing a schematic sketch of the expected local signatures of the bulk electronic structure caused by the scattering at single defects, the authors promise similar results as discussed in ref. [6]. The MS would gain much, if the authors could provide evidence for single-scattering events at single local defects and at the same time for interference patterns caused by the scattering at different defects. The single scattering events should be seen in the real space STM conductance map around the defect positions, whereas the interference patterns are visible in k-space by Fourier transformation as shown.

As the authors discussed the bulk electronic structure with respect to flat areas on the isoenergetic surfaces, it becomes obvious, that at certain energies, electrons are focussed in certain directions.

In the discussion of the electronic structure of the fcc material in contrast to the tetragonal symmetry superimposed by the (001) surface many inconsistencies about the lattice parameters and the lattice directions deserve careful checking. The directions (001), (010) etc have to be defined with respect to either the cubic or the tetragonal cell.

In figure 1, the STM topography is not clear to me, if it was measured in constant height or constant current mode. The square lattice expected at the (001) surface is not accessible. There are no scales for x, y, and z direction given. The z-position of the defects in the sketch is not clear. In panel c, the energy for the constant energy surface should be given explicitly. In panel e and f, the directions x and y are not clear.

In figure 2, give the band structure $e(k)$ also on the line Gamma-L, which is important for the understanding of the elongated shape of the electron lenses around the L point. Give more details on the calculation of the ARPES result. Which features from panel a are reflected in the theoretical ARPES result? What is the z scale in panels e and f?

How were the V-set and I-set values chosen for the different bias panels in fig. 2b)? The set point for the conductance maps in panels 2e and 2f should be given.

In figure 3, the surface periodicity in $[-\pi/a_{\text{tet}}, +\pi/a_{\text{tet}}]$ should be reflected in the $\rho(q)$ plot. Give details on the considered area of the Fourier transforms shown in panels a-h.

It should be clearly stated that the QPI pattern are in k-space. What is the lattice parameter here, cubic or tetragonal cell? What is the reason to choose a limit of $2\pi/a$? The ratio p_z/p_{xy} is not clear, the determination/optimization is not explained. Since the orbital character of a pure p-orbital

tight binding model is strongly connected to the x, y, z component of the k vector, this ratio gives the direction of the k vector, but not necessarily the direction of the wave packet propagation. The change of the p_z/p_{xy} ratio with energy is not convincing, even not by including the scattering at the cationic defects. Is there experimental evidence for the dominance of cationic defects?

The Green's functions in eq. 1 and 2 need check of signs and indices, I guess. The 2-dimensional character of the considered q vector and the connection to the 3-dimensional k space should be explained. To my impression, both equations 1 and 2 are more appropriate for the Supplement, since the general audience can not judge these equations. The reason for choosing a certain $V(E)$ as a perturbation of the Green's function should be given.

In figure 4a, the directions of the slices are not clear. The differences in panel b between unfolded and folded isoenergy surfaces are not expected. To my experience, a pure folding by lowering symmetry does not change the electronic eigenstates at all, just the positions in k space are shifted.

The definitions of the areas and the flatness should be given, and the connection to the in-plane propagation. How is the second direction for the determination of the isoenergy flatness treated? The red line at about $1.1 \pi/c$ is outside of the considered tetragonal Brillouin zone. The energy E for panel b should be given. The minimum of the area at the Brillouin zone boundary is questionable. The values for k_z large than π/c are redundant.

The q vectors defined in panel c deserve some more consideration. To my opinion, vectors q_1 and q_2 describe the same periodicity, considering the lattice periodicity of a_{tet} . Vector q_4 does not connect states of opposite travelling directions, so no interference is expected by scattering at different defects. The flatness of the connected areas has to be given more attention, since the flatness determines the strength of the wave packet and the intensity of the interference wave pattern. The energy for panel c should be given. In panel d, the energy gap located at the fcc L point is given at the tetragonal Z point, this is unexpected. Where is the vector q_3' located?

On page 15, the definition of the logarithmic derivative needs some more consideration. $V(V)$ is not obvious.

The choice of the scattering potential is not given. which type(s) of defects and which defect distribution were considered? Which energy dependence of $V_0(E)$ was chosen?

How is the vacuum decay at 5 Angstroms treated by a bulk like perturbed Green's function?

The $V_0(E)$ and $\rho_{xy}/\rho_z(E)$ are fitting parameters to obtain the surface density of states and the pseudo bulk electronic structure by $G(q,E)$. This should be explained.

In figure S2, x, y, and z scales should be given. Are the I panels constant height images? Why is the 4-fold symmetry not obvious in the Fourier transforms?

The energy for figure S3 should be given.

In figure S4, give the energy for the plots. Where are the q_2 and q_4 vectors located?

Best,

Peter Zahn

We thank all the referees for their time in reviewing our manuscript and for their constructive criticism. All referees suggest that our work should be published after revision. We have carefully considered the points raised by the reviewers and have revised our manuscript accordingly. We are confident that the modifications and our reply address all the points raised by the reviewers. We have copied below the reviewer reports with our replies indented, italic and in blue. Reference numbers, where provided, refer to the reference list in the revised manuscript.

Reviewer #1 (Remarks to the Author):

In "Tomographic mapping of the hidden dimension in quasiparticle interference" by Marques et al., the authors present a method for analyzing quasiparticle interference images, which permits extraction of information about the bulk electronic structure of galena, a fully three-dimensional--rather than quasi-two dimensional--material. They develop a method based on T-matrix calculations for extracting this information, in order to go beyond past work limited to study of quasi-two dimensional systems. Experimental results are supported by density functional theory and tight-binding theoretical methods to demonstrate the effectiveness of their approach.

The work serves as a useful step in studying truly three dimensional electronic structures, showing excellent agreement between experiment and theory. Given the widespread use of quasiparticle interference imaging in study of condensed matter systems, this proof-of-concept work addresses an important issue in extending this experimental technique to a broader class of materials. I therefore think is suitable for publication in Nature Communications. I do ask, however, for the authors to clarify how they determined the ratios of p_z and $p_{\{x,y\}}$ orbitals used to match to experiment for Fig. 3. Was this ratio determined manually and by eye, or was it automated and achieved by a quantitative fit? If this was achieved manually, the authors should acknowledge this, comment on the shortcomings or why it is good enough, and discuss how to make this more quantitative. If it was achieved quantitatively, explicit discussion of this approach should be added, again commenting on shortcomings and possible improvements.

The p_z/p_{xy} ratios shown in Figure 3 were found manually. There is no particular shortcoming in this procedure other than usual trial-and-error adjustments that fortunately can be done rather quickly using our method. We now clarify this point in the text (pg. 8, paragraph after eq. 2).

Reviewer #2 (Remarks to the Author):

In this manuscript, Marques et al. investigated the quasiparticle interference (QPI) images on PbS(100) surface through scanning tunneling microscopy. By considering the folded band structures in minimal supercell, they found the QPI images at different bias are dominated by the scattering vectors in energy-dependent kz -plane. Through T matrix calculations with full 3D electronic structure and orbital composition of surface band, the QPI patterns can be theoretically computed and accorded with experiments well. Their results suggest the 3D electronic structures have strong contribution in the QPI images, which can be used to extract the bulk band structures. This work is quite interesting, and the theoretical framework is thought to be convincing. However, before this manuscript is published, some revisions are needed. My comments and questions are as follow.

1. According to their theoretical framework, the QPI images are dominated by the energy-dependent kz plane, which can be derived as shown in Fig.4 for $E_b=0.85\text{eV}$ for example. The calculated QPI patterns at different energies are shown in Fig.3, where however the dominating kz planes for each energy were not given. The corresponding kz planes at each energy which are used to calculate the QPI patterns in Figure 3 should also be presented in these figures in order for the authors to better follow their theoretical framework.

The calculated QPI patterns in Figure 3 are obtained by integrating over all kz planes. To clarify this, we have added a short description in the revised manuscript on page 8 (immediately after eq. 2). We later discuss the dominant kz planes to provide an intuitive explanation of the origin of the main scattering vectors.

2. The authors show the scattering vectors connecting the pocket-like constant energy surface (CES) in Fig. 4c. Since the QPI patterns come from the energy-dependent kz plane, the CES in different kz plane should be quite different from others. I wonder how the scattering vectors connect the CES in other kz planes.

We refer the referee to figure 4a, which shows how the scattering vector changes for different kz planes and at different energies. As we show, only some kz planes contribute significantly. We have added a figure below to highlight this point, see fig. R1.

Figure R1: Sketch of cuts parallel to k_y , for constant energy surfaces based on the tight-binding model from Lent et al. [Superlattices and Microstructures **2**, 491-499 (1986)]. The spatial directions are in relation to the minimal Brillouin zone, represented by the grey rectangle. The dotted grey lines show the limits of the FCC Brillouin zone and the high symmetry points are indicated for comparison. We indicate dominant scattering vectors for a defect in the top surface layer. The red arrow indicates the dominant scattering vector and k_z plane seen in QPI. Once the pockets at the L-points connect and cross the $k_z=0$ plane, there are two dominant scattering planes, $k_z=0$ (blue arrow) and the scattering plane indicated by the red arrow.

3. The authors only show the dispersion of q_1 and q_3 in Fig. 4d. The scattering vectors q_2 and q_4 in folding BZ are missing. Why?

For the sake of clarity we have removed these two vectors in the revised manuscript, they are indeed hard to discern. Accordingly, we changed the label of q_3 to q_2 in Figure 4 of the main manuscript and Figure S5 in the supplementary.

Reviewer #3 (Remarks to the Author):

Dear editorial office, dear authors,

I read the MS 'Tomographic mapping of the hidden dimension in quasi-particle interference' by Marques, Wahl et al.

The presented work is very impressive to me, it's worth to be presented to a broad audience.

Still, I feel that the material and the discussion need quite some improvements

and changes following the questions raised below.
After major revisions, the manuscript might be suitable for publication.

As explained in very detail, the authors have found signatures of the bulk electronic structure on the surface of a crystal provided by the scattering at point-like impurities in the surface-near region. The density of states at the surface was mapped by differential conductance maps acquired by STM. So, long-range periodic changes of the density of states were found which are related to features of a perturbed bulk-like electronic structure which was derived by a Green's function technique. This relation of experimental and theoretical results is not very convincing since it is not clear how the modified bulk electronic structure reflects the surface electronic structure, since the surface electronic structure was not considered explicitly.

While it is true that the surface electronic structure is not considered explicitly, we note that this is justified by the ARPES results in fig. 2c, which show excellent agreement with the bulk electronic structure.

By showing a schematic sketch of the expected local signatures of the bulk electronic structure caused by the scattering at single defects, the authors promise similar results as discussed in ref. [6]. The MS would gain much, if the authors could provide evidence for single-scattering events at single local defects and at the same time for interference patterns caused by the scattering at different defects. The single scattering events should be seen in the real space STM conductance map around the defect positions, whereas the interference patterns are visible in k-space by Fourier transformation as shown.

Fig. 2e, f show the real-space conductance modulations around defects, where the single scattering events around defects can be clearly seen. We note that we cannot control the number and position of defects because they arise from the natural impurities in the mineral sample.

As the authors discussed the bulk electronic structure with respect to flat areas on the isoenergetic surfaces, it becomes obvious, that at certain energies, electrons are focussed in certain directions.

In the discussion of the electronic structure of the fcc material in contrast to the tetragonal symmetry superimposed by the (001) surface many inconsistencies about the lattice parameters and the lattice directions deserve careful checking. The directions (001), (010) etc have to be defined with respect to either the cubic or the tetragonal cell.

All lattice parameters are defined with respect to the conventional unit cell as shown in fig. 1b on the right (“minimal cell”). This is the lattice observed in the STM images and differential conductance maps. The Bragg peaks that appear in the Fourier transformation correspond to $2\pi/a$, where a is the distance between Pb atoms, and is the lattice constant both of the bulk FCC unit cell as well as along [100] and [010] of the tetragonal unit cell. We have added a coordinate cross in fig. 1b for clarity, and mention this now explicitly in the caption of fig. 1.

In figure 1, the STM topography is not clear to me, if it was measured in constant height or constant current mode.

The topography was measured in constant current mode, we have added a corresponding comment in the caption. We have added the 2D image corresponding to the STM topography of Fig. 1 in the supplementary, together with its Fourier transform, in Fig. S1 in the new section S2. We have also added a description of the imaged atomic lattice.

The square lattice expected at the (001) surface is not accessible. There are no scales for x, y, and z direction given. The z-position of the defects in the sketch is not clear. In panel c, the energy for the constant energy surface should be given explicitly. In panel e and f, the directions x and y are not clear.

See above, the new supplementary fig. S1 has a scale bar and shows the atomic lattice in the Fourier transformation.

Panel c is just a schematic of a Fermi surface of a cubic system for illustration, so we cannot provide an energy. We have clarified that in the caption.

In figure 2, give the band structure $e(k)$ also on the line Gamma-L, which is important for the understanding of the elongated shape of the electron lenses around the L point. Give more details on the calculation of the ARPES result. Which features from panel a are reflected in the theoretical ARPES result?

The revised Figure 2 now includes the Γ -L direction as well. The ARPES measures the surface projection of the bulk electronic structure, over a Lorentzian distribution in k_z around a central value set by the photon energy, as described in detail in the Supplementary information.

Therefore, multiple features of the band structure in (a) contribute to the experimentally measured electronic structure by ARPES, necessitating the explicit simulation of this from the DFT calculations, which we show besides the ARPES in Fig. 2c.

What is the z scale in panels e and f?

Panels e and f are differential conductance (dI/dV) maps, so do not represent a z-scale. We have added values to the colour bars for clarity.

How were the V-set and I-set values chosen for the different bias panels in fig. 2b)? The set point for the conductance maps in panels 2e and 2f should be given.

The choice of the set point parameters, bias voltage V and current I , for the data in Figure 2b is explained in the supplementary in section S3 and Figure S2. We have added a sentence about the reasoning behind the choice of parameters. We have also added the setpoint values for the conductance maps in the supplementary material in section S3, Table S2.

In figure 3, the surface periodicity in $[-\pi/a_{\text{tet}}, +\pi/a_{\text{tet}}]$ should be reflected in the $\rho(q)$ plot. Give details on the considered area of the Fourier transforms shown in panels a-h.

The surface periodicity in the $\rho(q)$ plots does not appear because they are calculated using a continuum model, where $\rho(q)$ is not periodic in the unit cell (see, e.g., Kreisel et al., PRL 114, 217002; Choubey, et al., PRB 96, 174523). The lateral size of the dI/dV maps used in Figure 3 were included in the supplementary in Table S2 as Δx .

It should be clearly stated that the QPI pattern are in k-space. What is the lattice parameter here, cubic or tetragonal cell? What is the reason to choose a limit of $2\pi/a$?

The reason to show the QPI maps up to $2\pi/a$ is that this is usually the range in which QPI is observed. The lattice Green's function is periodic in $2\pi/a$, when considering the corresponding continuum Green's function with realistic modelling of the overlap with the tip of the STM, the QPI signal is usually governed by contributions within the range up to $2\pi/a$. A practical reason for choosing a limit of $2\pi/a$ for the measurements (and not something smaller) is that this ensures that the atomic peaks are not aliased and can be used to correct for drift occurring over the course of the measurement, as discussed in the supplementary (section S4). The lattice parameter is the one corresponding to the minimal supercell, a , corresponding to the Pb square lattice, but we note that it is the same in the primitive cell.

The ratio p_z/p_{xy} is not clear, the determination/optimization is not explained. Since the orbital character of a pure p-orbital tight binding model is

strongly connected to the x, y, z component of the k vector, this ratio gives the direction of the k vector, but not necessarily the direction of the wave packet propagation. The change of the p_z/p_{xy} ratio with energy is not convincing, even not by including the scattering at the cationic defects.

In our tight-binding model, the x, y and z components of the p orbitals have been aligned with the [100], [010] and [001] lattice directions of the minimal tetragonal supercell shown in Figure 1, respectively. Accordingly, the z component is parallel to the surface normal [001], and the other two components are parallel to the surface plane. Within this framework, we have then varied the p_z/p_{xy} ratio until the resulting QPI pattern shows the best agreement with the corresponding experimental data at each bias potential.

Is there experimental evidence for the dominance of cationic defects?

As discussed in the supplementary material, the sample is argentiferous galena, i.e. silver-rich, with silver and bismuth being the most prevalent defects as confirmed by the elemental analysis shown in supplementary section S1/table S1. Both are cationic defects.

The Green's functions in eq. 1 and 2 need check of signs and indices, I guess. The 2-dimensional character of the considered q vector and the connection to the 3-dimensional k space should be explained. To my impression, both equations 1 and 2 are more appropriate for the Supplement, since the general audience can not judge these equations. The reason for choosing a certain $V(E)$ as a perturbation of the Green's function should be given.

We have fixed the sign in equation 1 and indices in equation 2. For both k and q vectors, we have used the same coordinate system based on the lattice parameters of our minimal tetragonal supercell shown in Figure 1. Accordingly, both k_x and q_x (k_y and q_y) are along the conventional 100 (010) crystalline direction in PbS, and k_z is parallel to 001. We would like to keep these equations in the main text, as they are crucial to our discussion of the impurity scattering and their orbital and bias dependencies.

In figure 4a, the directions of the slices are not clear. The differences in panel b between unfolded and folded isoenergy surfaces are not expected. To my experience, a pure folding by lowering symmetry does not change the electronic eigenstates at all, just the positions in k space are shifted. The definitions of the areas and the flatness should be given, and the connection to the in-plane propagation. How is the second direction for the determination of the isoenergy flatness treated? The red line at about $1.1 \pi/c$

is outside of the considered tetragonal Brillouin zone. The energy E for panel b should be given. The minimum of the area at the Brillouin zone boundary is questionable.

We agree that zone folding by itself does not have any impact on the electronic structure. However, it can allow mapping out the dominant bulk q vectors that contribute the most to the observed QPI at the surface. This mapping may not be of much utility if the given system has a two-dimensional electronic structure. For three-dimensional systems like PbS, this is entirely different. In such cases, all three components of the impurity's q vector, including q_z , need to be considered to account for an observed QPI pattern. Unlike the q_x and q_y components, finding a dominant q_z component is not an easy task. It requires a precise mapping between the electronic bands in the bulk Brillouin zone and their projections at the surface. Our formalism provides such a mapping. As discussed in the manuscript, the zone folding associated with a surface termination defines which q vectors dominate QPI intensities at a given bias potential. They are the q vectors whose q_z components match kz -planes in which the primitive Fermi surface and its folded image form the largest cross-sectional area. In other words, for that particular q_z , an impurity can strongly scatter the wave vector k to $k+q$. So all one needs to do is consider a properly folded bulk electronic structure, search for those kz planes that exhibit the flattest bands, and then integrate their contributions. We have added the energy for panel 4b in the caption.

The values for k_z large than π/c are redundant. The q vectors defined in panel c deserve some more consideration. To my opinion, vectors q_1 and q_2 describe the same periodicity, considering the lattice periodicity of a_{tet} . Vector q_4 does not connect states of opposite travelling directions, so no interference is expected by scattering at different defects. The flatness of the connected areas has to given more attention, since the flatness determines the strength of the wave packet and the intensity of the interference wave pattern. The energy for panel c should be given. In panel d, the energy gap located at the fcc L point is given at the tetragonal Z point, this is unexpected. Where is the vector q_3' located?

Not only states with opposite travelling direction contribute to QPI (see, e.g., the famous octet model for the cuprate superconductors), however we have removed the vector as it is difficult to see in the calculated QPI for clarity.

We have added a section in the supplementary (new section S7) and Fig. S7 to explain the distinction between q_3 and q'_3 . The difference between these two scattering vectors arises due to the change in topology from disconnected pockets close to the top (bottom) of the valence (conduction) bands, where the dominant q -vector is q_3 , to connected pockets at the energy $M1$ (maximum/minimum in the band structure along Γ -K in the FCC Brillouin zone) where q_3 becomes zero and is replaced by q'_3 .

On page 15, the definition of the logarithmic derivative needs some more consideration. $V(V)$ is not obvious.

We realize that the notation with the whole equation on one line was not very clear. In the revised version we have typeset the equation on its own line (pg. 15), which we are confident removes any ambiguity.

The choice of the scattering potential is not given. which type(s) of defects and which defect distribution were considered? Which energy dependence of $V_0(E)$ was chosen?

As discussed in the manuscript, we have assumed the impurities are non-magnetic and randomly distributed in the whole system. Accordingly, we can describe them by a spin- and momentum-independent potential $V=V_0I$, where V_0 is a constant and I is a unity matrix. Also, V_0 can be fixed for all calculations as it does not affect the details of the resulting QPI patterns. We have added details of the potential in the methods section, near the top of pg. 16.

In our calculations, we do not choose the type of defects directly. As mentioned above, we treat the existing scatters by a mean-field potential which is constant throughout the reciprocal space. Instead, by changing the p_z/p_{xy} ratio of conduction Pb and valence S bands, we find the optimal configuration reproducing the experimental QPI pattern at a given bias potential. This enables us to determine what types of impurity bound states within which energy window are likely responsible for the observed QPI features.

How is the vacuum decay at 5 Angstroms treated by a bulk like perturbed Green's function?

We use atomic-like Wannier functions with well-defined spatial distributions for the Bloch wave functions for the projection onto a tight-binding model and to determine the vacuum decay. This allows us to perform a basis transformation from the lattice model to the continuum model, as described in [Choubey, et al., PRB 96, 174523 (2017); Kreisel et al., PRL 114, 217002], References 22 and 23 in the revised manuscript. In this way, we can map the bulk QPI onto any plane in the continuum space. One possible choice is the plane within which the STM tip scans the surface. In our calculations, we assume that it is $z=5 \text{ \AA}$ above the topmost PbS atomic layer in the tetragonal supercell.

The $V_0(E)$ and $\rho_{xy}/\rho_z(E)$ are fitting parameters to obtain the surface density of states and the pseudo bulk electronic structure by $G(q,E)$. This should be explained.

As explained earlier, V_0 is just a constant in our calculations. The only fitting parameter in our formalism is the p_z/p_{xy} orbital ratio.

In figure S2, x, y, and z scales should be given. Are the I panels constant height images? Why is the 4-fold symmetry not obvious in the Fourier transforms?

[this refers to fig. S3 of the revised manuscript] We have added scale bars and coordinate axis to the real-space panels of Figure S3. The I (current) panels are the current images recorded simultaneously with the dI/dV maps, which are spectroscopic maps recorded with open feedback loop while recording the tunnelling spectrum, but closed feedback loop while moving the tip to the next point of the image at the setpoint bias/current (see table S2). It is these current images which are then used to calculate $d\ln I/d\ln V$.

The Fourier transforms do not appear 4-fold symmetric due to piezo creep and thermal drift, which are a consequence of the dI/dV maps being taken over a period of 24 hours and at a temperature of 20K. This is responsible for a noticeable distortion in the real-space dI/dV images, which is reflected in the Fourier transform, where the two sets of Bragg peaks appear to not correspond to the same lattice constant and the angle between them is not exactly 90 degrees. To correct for this, we perform the process described in section S4.

The energy for figure S3 should be given.

[fig. S4 of the revised manuscript] While we think it is irrelevant for what the figure is trying to explain (namely how the data processing affects what is shown), we have added the energy.

In figure S4, give the energy for the plots. Where are the q_2 and q_4 vectors located?

[this refers to fig. S5 of the revised manuscript] We do not understand this comment. The graphs are as a function of bias voltage/energy, there is no other energy.

Best,
Peter Zahn

List of changes:

- *Fig. 1, Added crystallographic directions and definition of lattice parameters in fig. 1b for the tetragonal unit cell*
- *Caption of fig. 1: added statement that the manuscript uses a notation referring to the tetragonal supercell throughout, defined lattice parameters of the unit cell; made clear that fig. 1c is a schematic Brillouin zone*
- *Pg. 8, added reference to method section for the Green's function, further added clarification below eq. 2 how the QPI signal is calculated. Added sentence to clarify that orbital ratio is adjusted manually.*
- *Pg. 9: clarified reference to the in-plane directions $[10]$ and $[01]$ in the discussion of fig. 3*
- *Fig. 3: modified coordinate cross in first panel to only include the in-plane direction, consistent with notation used in the text*
- *Fig. 4/caption of fig. 4: changed error bars to represent q -space resolution of the map (also fig.*
- *Method section, pg. 16, added a sentence on how the impurity potential is defined, as well as details on how the continuum QPI is calculated from Wannier functions.*
- *Fixed a few typos throughout the text, and reference to the supplementary material*
- *Added data and code availability statements*
- *Added references 22, 23*

Supplementary:

- *Added section S2/fig. S1 to explain the topography and show the atomic resolution*

- *Section S3, added a statement on the tunnelling matrix element to rationalize the choice of setpoint conditions. Added table S2 which lists the setpoint conditions used for measurements shown in the main text.*
- *Caption of fig. S4 (former fig. S3): added bias voltage for the QPI data shown.*
- *Added section S7/fig. S7 to explain the different between q_2 and q_2^{\prime} .*

REVIEWER COMMENTS

Reviewer #1 (Remarks to the Author):

I am satisfied with the changes made by the authors and think the manuscript can now be published in Nature Communications.

Reviewer #2 (Remarks to the Author):

I think the referee comments have been properly addressed and the manuscript is ready for publication now.

Dear editors, dear authors,

This is my 2nd report on the MS 'Tomographic Mapping of the hidden dimension in QPI' by Marques et al.

I appreciate that the authors improved and extended the article, as well as the SI.

I'm completely convinced that the experimental results are correct and an impressive example of bulk electronic structure investigations by having access to the surface only.

I feel that the connection of theoretical model and experimental results is not explained well enough.

So, I still have some questions and remarks to the authors which should be answered before publication.

In the discussion, the perturbed density is considered for q vectors in the plane (q_x, q_y) only.

The statement on k_z summation does not solve this issue.

To my opinion, all q vectors with a fixed q_x and q_y , differing by the q_z value, contribute to the visible perturbations of the density at the surface. The strength of the influence is determined by the reflection properties of the surface potential to the specific (q_x, q_y, q_z) density waves.

This might be covered to a certain extent by the ratio p_z/p_{xy} introduced by the authors to weight the different orbital contributions. I guess that due to the strong coupling of orbital character to the q -vector direction in a pure p -orbital tight-binding model, this ratio fixes the angle between q vector and surface normal to a certain range.

Concerning fig. 4, I have the following remarks.

The formalism presented allows the mapping of the bulk electronic structure to the symmetry imposed by the considered surface facet.

The visibility of the QPI at the surface is given by the reflection of the perturbed bulk states (by the impurity) at the surface potential which differs a lot from the bulk potential. These different scattering amplitudes might be reflected to a certain extent by the p_z/p_{xy} ratios.

The given explanation of the stationary q vectors is not correct in my opinion. What you have to look for are parallel constant-energy surface areas which might include q -vectors with non-zero z -component. What you measure at the surface is (q_x, q_y) only, but q_z is diminished by the reflection at the surface. The consideration of the surface as a truncated bulk region and expanding the (perturbed) Bloch states in this vacuum region seems a quite rough approximation. Please comment on this.

The surface area shown in panel 4b is not correct to my opinion, 1/ from symmetry reasons there should be zero slope at $k_z=\pi/c$, as it is in the unfolded case,

2/ the folded area should be twice the unfolded one from symmetry reasons.

I suppose that the nested constant energy pockets around the fcc-L point are not treated correctly.

The flatness definition based on the area in (k_x, k_y)-plane should be given explicitly since it is not standard. There should be two types of flatnesses defined for (100) and (110) directions. What you have determined are stationary q -vectors in the (q_x, q_y)-plane. I guess there are more with stronger nesting properties. In the attached PNG figure I draw some arrows indicating positions of nesting q -vectors which connect iso-energy surface

parts which nest much better than the ridge-like areas you mark for $E = -1.2\text{eV}$ and $E = -1.75\text{eV}$. Your choice might be caused by the used definition of flatness. To my opinion, the restriction to $(q_x, q_y, q_z = 0)$ vectors is not sufficient. To measure a QPI from sub-surface defects a stationary q -vector with non-zero q_z is mandatory. You find such nesting q -vectors for $E_b = -0.85\text{eV}$, -1.2eV , but not for $E = -1.75\text{eV}$, -2.2eV , see arrows in the attached PNG file.

The nesting condition for vector q_2 seems to be very weak, q_2' seems reasonable.

When you consider the perturbed density $\rho(q_x, q_y, q_z, E)$, the q vector does not describe the impurity, since you consider exactly one impurity site with a delta-like perturbation given by the strength of 0.1V . The unit Volt seems not correct. Please reconsider the unit of the impurity potential strength.

For fig. 4d, please give the complete energy-dependence of the orbital contribution ratio p_z/p_{xy} which is given for selected energies in fig. 3. In fig. 4c the vectors q_1 and q_2 have to be given more clearly. The panels 4c, S7b, c, and d puzzle me a lot, since the cuts do not fit together considering similar energies. In figure caption 4f, there is confusion about V_b and E_b .

Concerning table S2 my question is, why you have chosen I_{set} and V_{set} as it was done.

It is not obvious why the tip is positioned lower or higher for the different voltage ranges measured.

Why was a positive V_{set} chosen to measure at negative bias values?

The ranges $-0.9 \dots -1$ and $-1.1 \dots -1.6\text{V}$ seem to be measured with identical parameters. May you can improve the table.

When describing the model formalism, eq. 4 is redundant to eq. 2 in my understanding. If I'm wrong, please let me know the reason.

My last question in report one which you did not get, was a bit screwed up - sorry.

The first part pointed to fig. S3 which was answered and changed in the figure.

The second one concerned fig. S4 which was solved by dropping vectors q_2 and q_4 from the discussion.

It is a nice piece of work which needs a bit more polishing.

Best regards,
Peter Zah

We thank Prof Zahn for his time in reviewing our manuscript, his diligent review and for his constructive criticism, highlighting that our manuscript represents “a nice piece of work which needs a bit more polishing”. We have carefully considered the points raised and revised our manuscript accordingly. We are confident that our reply and the modifications of the main text address the points raised by the referee. We have copied below the report by Prof Zahn with our replies indented, italic and in blue. Reference numbers refer to the reference list in the revised manuscript. Our comments are typeset in blue, italic and indented.

Dear editors, dear authors,

This is my 2nd report on the MS 'Tomographic Mapping of the hidden dimension in QPI' by Marques et al.

I appreciate that the authors improved and extended the article, as well as the SI.

I'm completely convinced that the experimental results are correct and an impressive example of bulk electronic structure investigations by having access to the surface only.

I feel that the connection of theoretical model and experimental results is not explained well enough. So, I still have some questions and remarks to the authors which should be answered before publication.

In the discussion, the perturbed density is considered or q vectors in the plane (q_x, q_y) only. The statement on k_z summation does not solve this issue. To my opinion, all q vectors with a fixed q_x and q_y , differing by the q_z value, contribute to the visible perturbations of the density at the surface. The strength of the influence is determined by the reflection properties of the surface potential to the specific (q_x, q_y, q_z) density waves.

This might be covered to a certain extent by the ratio p_z/p_{xy} introduced by the authors to weight the different orbital contributions. I guess that due to the strong coupling of orbital character to the q -vector direction in a pure p-orbital tight-binding model, this ratio fixes the angle between q vector and surface normal to a certain range.

We completely agree with the reviewer that the observed QPI at the surface $\tilde{\rho}(\tilde{q}_x, \tilde{q}_y)$ results from all possible bulk q vectors sharing the same (q_x, q_y) components. The additional q_z component defines the extent by which the perturbed bulk bands contribute to the joint surface density of states. As the reviewer correctly put it, the surface potential, due to its symmetry constraints, acts as a reflector selectively picking up the bulk $\rho(q_x, q_y)|_{k_z}$ at different k_z planes and mixing them up to form an observable STM QPI $\tilde{\rho}(\tilde{q}_x, \tilde{q}_y)$. We somehow missed clarifying this in our discussions related to Eq. (1) in the previous revision. To avoid any confusion, we have now explicitly stated in the paragraph below the equation (1) on page 8 that in our formalism, we first calculate $\rho(q_x, q_y)|_{k_z}$ for all the possible k_z planes in the folded Brillouin zone of the bulk PbS and then integrate them to construct the surface QPI $\tilde{\rho}(\tilde{q}_x, \tilde{q}_y)$.

The reviewer is also correct that this effect is partly reflected by the change in the p_z/p_{xy} ratio at different energies, as the directional polarity of the p orbital is strongly coupled with the scattering q vectors. This is also clarified in the same paragraph in revised manuscript.

Concerning fig. 4, I have the following remarks.

The formalism presented allows the mapping of the bulk electronic structure to the symmetry imposed by the considered surface facet.

The visibility of the QPI at the surface is given by the reflection of the perturbed bulk states (by the impurity) at the surface potential which differs a lot from the bulk potential. These different scattering amplitudes might be reflected to a certain extent by the p_z/p_{xy} ratios. The given explanation of the stationary q vectors is not correct in my opinion. What you have to look for are parallel constant-energy surface areas which might include q -vectors with non-zero z -component.

What you measure at the surface is (q_x, q_y) only, but q_z is diminished by the reflection at the surface. The consideration of the surface as a truncated bulk region and expanding the (perturbed) Bloch states in this vacuum region seems a quite rough approximation. Please comment on this.

As explained above, we do follow the same step as those suggested by the reviewer for calculating the surface QPI. We apologise if our previous revision was not clear enough on this point. We hope our new corrections remove this confusion.

The surface area shown in panel 4b is not correct to my opinion,
 1/ from symmetry reasons there should be zero slope at $k_z = \pi/c$, as it is in the unfolded case,
 2/ the folded area should be twice the unfolded one from symmetry reasons. I suppose that the nested constant energy pockets around the fcc-L point are not treated correctly.

(1) *The Fermi pocket centred at the L point $(\frac{\pi}{a}, \frac{\pi}{a}, \frac{\pi}{a})$ has a strong energy dispersion as expected from a bulk 3D system. Thus, the only k -points at which it can have a zero slope are the high symmetry k -points such as L point. Everywhere else, the band dispersions should have a finite slope. To demonstrate this, below, we have shown the electronic band structure of PbS along W-L-W in the unfolded BZ, see Figure R1-(a). This direction corresponds to $[100]$ axis in the folded zone with $k_z = \pi/a$. As can be seen, both conduction and valence bands show a dispersion with a finite slope at k -points away from the L point. Figure R1-(b) displays this behaviour more clearly. Here, the most noticeable feature is the maximum slope achieved at binding energy $E \sim -0.85$ eV even though $k_z = \pi/a$. Thus, we can expect that as long as the bias potential is strong enough to cross the energy bands away from their extrema, the overall flatness of the resulting energy pockets should be finite.*

Figure R1. (a) Electronic band dispersions of top valence band (red) and bottom conduction band (green) along W-L-W direction, corresponding to $[100]$ axis in the folded space with $k_z = \pi/a$. The horizontal dashed line corresponds to the binding energy $E_B = -0.85$ eV. (b) The calculated slope for each band along W-L-W. The open and filled black dots denote two possible slopes of the valence band at $E_B = -0.85$ eV.

(2) *the total area of the folded and combined energy pockets cannot be twice the area of the unfolded one, as there is an overlap between the original Fermi surface and its folded image. To demonstrate this, we show in Figure R2, the Fermi surface in the unfolded and folded BZs at $E = -0.85$ eV. Comparing Figures R2-(a) and (b), one can notice that the folding has a reflection effect, causing an overlap between the original energy pocket (shown in orange) with its folded image (shown in black). Focusing only on one of the pockets, we can see that there is a significant overlap between the two sections, creating a large common area (See Figure R2-(c) and (d)). Accordingly, only the non-overlapping part has a double contribution*

Figure R2. The Fermi surface calculated for (a) unfolded BZ and (b) folded BZ at $E_B = -0.85$ eV. (c) The magnified front view of one of the energy pockets (in orange) and its folded image (in black). (d) The cross-sectional area resulting from the k_z cut shown with dashed red line in (c).

which after being added to the common area amounts to the total cross-sectional area of the Fermi surface in the folded BZ. We have added a statement in the caption of fig. 4 (pg. 11) to address this point.

We realize that the definition of flatness and its discussion may have been confusing. To address this, we have modified the definition to state that planes where $\left(\frac{\partial A}{\partial k_z}\right)^{-1}$ is maximized dominate the QPI. There are two contributing factors to the QPI intensity: (1) The flatness of the original band, as, e.g., expressed by the effective mass – the flatter a band locally, the higher its contribution to the QPI signal due to an increased joint density of states. (2) Bands with a group velocity parallel to the direction from the tip to the scatterer will contribute more.

While (1) is directly linked to the cyclotron effective mass is defined as $m^* = \frac{\hbar^2}{2\pi} \frac{\partial A}{\partial E}$, i.e., it is proportional to the first derivative of the cross-sectional area with respect to energy (see, "Principles of the theory of solids" by J. M. Ziman, Cambridge University Press (1972)), (2) can be linked to extremal orbits in a plane orthogonal to the surface, if one assumes that scattering is dominated by surface defects. Extremal orbits, again as is usually done to identify the dominant cross sections through the Fermi surface for quantum oscillations, are determined by the condition that $\frac{\partial A}{\partial k_z} = 0$, where the z-direction here is normal to the surface. This is

equivalent to saying that planes contribute where $\left(\frac{\partial A}{\partial k_z}\right)^{-1}$ is maximized. We have changed the discussion in the manuscript to this second argument which we think is easier to follow (page 10), and updated the graph in fig. 4b (just to note, for $k_z=1$ (in units of π/c_0), the slope is not well-defined). We note that fig. 4b only provides an intuitive argument to understand why certain k_z planes dominate the QPI signal, the calculation accounts for all k_z planes.

What you have determined are stationary q-vectors in the (qx, qy)-plane. I guess there are more with stronger nesting properties. In the attached PNG figure I draw some arrows indicating positions of nesting q-vectors which connect iso-energy surface parts which nest much better than the ridge-like areas you mark for $E=1.2$ eV and $E=-1.75$ eV. Your choice might be caused by the used definition

of flatness. To my opinion, the restriction to $(q_x, q_y, q_z=0)$ vectors is not sufficient. To measure a QPI from sub-surface defects a stationary q -vector with non-zero q_z is mandatory. You find such nesting q -vectors for $E_b = -0.85eV, -1.2eV$, but not for $E = -1.75eV, -2.2eV$, see arrows in the attached PNG file. The nesting condition for vector q_2 seems to be very weak, q_2' seems reasonable.

We appreciate the reviewer for raising this important point which complements his first two comments. While the reviewer is in principle correct that the nesting vector he draws provides a stronger nesting, at the same time this wave vector has a group velocity with a large out-of-plane component, and thus is only relevant for defects which are comparatively deep in the sample. This means that the QPI due to this wave vector, when it reaches the surface, is already significantly weaker compared to that of surface defects unless there is a focussing effect as, e.g., found for bulk copper (see ref. 6 of our manuscript). In the general case though, the QPI will be dominated by surface defects and scattering vectors with a group velocity parallel to the surface. These scattering vectors are exactly the ones which dominate in the k_z plane identified by our flatness analysis in the folded zone. By symmetry, the dominant scattering vectors with in-plane group velocity and non-zero q_z (connecting planes with $k_z \neq 0$ of the cubic Brillouin zone) are mapped into the same plane, so are captured by our formalism, see fig. R3. So in principle, the scattering vector drawn by Prof Zahn is also accounted for in our calculations (see fig. R3): Due to the folding, the nesting vector is mapped into a plane with $q_z = 0$.

Figure R3. Comparison of a nesting vector q in the unfolded BZ and its counterpart q' in the folded BZ. These two nesting vectors, even though having different z components, can result in the same surface nesting vector \tilde{q} . More importantly q' is parallel to \tilde{q} , indicating that the observed $\rho(\tilde{q})$ can be reproduced by integrating $\rho(q)$ over parallel k_z planes in the folded BZ.

When you consider the perturbed density $\rho(q_x, q_y, q_z, E)$, the q vector does not describe the impurity, since you consider exactly one impurity site with a delta-like perturbation given by the strength of $0.1V$. The unit Volt seems not correct. Please reconsider the unit of the impurity potential strength.

We thank the reviewer for pointing out this. As the energy units usually used to analyze tunnelling spectra are in eV, and convert 1:1 from the bias voltage, we forgot the "e" for eV here. We have corrected that in the revised manuscript (pg. 16).

For fig. 4d, please give the complete energy-dependence of the orbital contribution ratio pz/pxy which is given for selected energies in fig. 3.

We have now supplied this in the Supplementary Information, fig. S8/Supplementary Section S9.

In fig. 4c the vectors q1 and q2 have to be given more clearly.

We have changed the line thickness and colors of the vector.

The panels 4c, S7b, c, and d puzzle me a lot, since the cuts do not fit together considering similar energies.

As mentioned in the caption of fig. S7, the band structure shown there is based on the tight-binding model of reference 2 in the supplementary material (Lent et al.), so there will be differences compared to the band structure shown in fig. 4c, which is based on the DFT model used throughout the main text.

In figure caption 4f, there is confusion about V_b and E_b .

Whenever we refer to measured data we provide the voltage at which the measurement was done because that is the parameter controlled in the experiment (here V_b). The energy and voltage are related by $E_b=eV_b$. Figure 4f refers to a measurement, so the bias voltage $V_b=-0.85V$ is provided, corresponding to an energy $E_b=-0.85eV$.

Concerning table S2 my question is, why you have chosen I_{set} and V_{set} as it was done. It is not obvious why the tip is positioned lower or higher for the different voltage ranges measured.

As we thought we explained in the revised supplementary section S3, the choice of different V_{set} and I_{set} values has purely technical reasons. Because of the rapid increase of the current with increasing voltage, if one tries to use the same I_{set} and V_{set} values, either the signal at small voltages becomes very small if it is adjusted to capture the QPI at larger voltages or, if one optimizes the settings for the signal at small voltages, the current amplifier saturates at larger voltages. Therefore, we have used different setpoint voltages and currents to capture the QPI signal in different voltage ranges, but have verified that our specific choice does not alter the result by checking that tunnelling spectra recorded with different setpoint conditions within the range shown remain identical.

Why was a positive V_{set} chosen to measure at negative bias values? The ranges -0.9 ..-1 and -1.1 .. -1.6V seem to be measured with identical parameters. May you can improve the table.

We tend to use setpoint parameters with a positive bias voltage. For some materials, it is easier to stabilize the tunnelling junction with a certain bias polarity. Here, stabilizing at positive bias voltages works better because there is a higher density of states (because the material is n-doped).

When describing the model formalism, eq. 4 is redundant to eq. 2 in my understanding. If I'm wrong, please let me know the reason.

In the revised manuscript, we have removed Eq. (2) from the main text.

My last question in report one which you did not get, was a bit screwed up- sorry.
The first part pointed to fig. S3 which was answered and changed in the figure.
The second one concerned fig. S4 which was solved by dropping vectors q_2 and q_4 from the discussion.

No problem, actually we think removing those vectors helped our discussions.

It is a nice piece of work which needs a bit more polishing.

We again thank the reviewer for his valuable comments. We hope with these changes, we have been able to address them appropriately, and he now finds our manuscript suitable for publication in Nature Communications.

Best regards,
Peter Zah

REVIEWERS' COMMENTS

Reviewer #3 (Remarks to the Author):

Dear editor, dear authors,

This is my 3rd report on the MS 'Tomographic mapping of the hidden dimension ..' Sorry for the long delay to answer your reply. In summary, I agree with the replies from the authors, and recommend the MS for publication.

The following three comments should be easily answered by the authors.

As they pointed out, the theoretical discussion of the QPI is restricted to scattering vectors q parallel to the surface, which is supported by strong arguments on the scattering of the density perturbations by the defects close to the surface.

This restriction of the theoretical considerations to $q_z=0$ has to be stated in the manuscript explicitly. The opposite is stated in the updated version below equation 1.

The discussion on the main contribution to the QPI concerning k_z position and the determination of the nesting vectors q_i is a bit weak to my experience. The definition of the flatness as shown in fig. 4b is connected to the effective cyclotron mass, which might differ from the effective mass connected to the group velocity and the density of states. Even more, the nesting condition for q -vectors is closely linked to the local flatness of the constant-energy surface at the positions connected by that q -vector. I don't feel, that this property can be well described by an integral value over the cross section of the combined electron pockets.

The last criticism concerns the usage of the 0.1 V or 0.1 eV as strength of the delta-like scattering potential.

The amplitude of a 3D delta-like scattering potential has to be of unit energy times volume, since the integral over the Wigner-Seitz volume of the scattering site is of this unit. It might be that the Wigner-Seitz cell volume is assumed here, which seems reasonable. If the authors agree they should add a remark in MS or Supplement.

Best,

Peter Zahn

We thank the referee for his positive assessment of our manuscript.

Dear editor, dear authors,

this is my 3rd report on the MS 'Tomographic mapping of the hidden dimension ..'

Sorry for the long delay to answer your reply.

In summary, I agree with the replies from the authors, and recommend the MS for publication.

The following three comments should be easily answered by the authors.

As they pointed out, the theoretical discussion of the QPI is restricted to scattering vectors q parallel to the surface, which is supported by strong arguments on the scattering of the density perturbations by the defects close to the surface.

This restriction of the theoretical considerations to $q_z=0$ has to be stated in the manuscript explicitly. The opposite is stated in the updated version below equation 1.

We have modified the text below eq. 1 to clarify this point, stating that scattering patterns are calculated in planes with $q_z=0$ but that vectors with non-zero q_z are accounted for through the folding.

The discussion on the main contribution to the QPI concerning k_z position and the determination of the nesting vectors q_i is a bit weak to my experience. The definition of the flatness as shown in fig. 4b is connected to the effective cyclotron mass, which might differ from the effective mass connected to the group velocity and the density of states. Even more, the nesting condition for q -vectors is closely linked to the local flatness of the constant-energy surface at the positions connected by that q -vector. I don't feel, that this property can be well described by an integral value over the cross section of the combined electron pockets.

The argument is only required to illustrate the dominant kz plane, but that the calculation integrates for all kz values, so does actually account for the details of the group velocity and density of states as mentioned by the referee. For a weakly correlated system as we consider here, the cyclotron orbit does provide a reasonable measure though to determine the dominant kz plane.

The last criticism concerns the usage of the 0.1 V or 0.1 eV as strength of the delta-like scattering potential. The amplitude of a 3D delta-like scattering potential has to be of unit energy times volume, since the integral over the Wigner-Seitz volume of the scattering site is of this unit. It might be that the Wigner-Seitz cell volume is assumed here, which seems reasonable. If the authors agree they should add a remark in MS or Supplement.

The potential is the change in on-site energy of the respective orbital. As such, it is normalized over the volume of the orbital wave function, which is usually larger than the Wigner-Seitz volume. We have modified the methods section to clarify this point.

Best, Peter Zahn